# ChromaFactor: Deconvolution of single-molecule chromatin organization with non-negative matrix factorization

**Laura M. Gunsalus**[1,2], **Michael J. Keiser**[2,3,4,5,6], **Katherine S. Pollard**[1,2,7,8,*]

**1** Gladstone Institute of Data Science & Biotechnology, Gladstone Institutes, San Francisco, California, United States of America, **2** Bakar Computational Health Sciences Institute, University of California, San Francisco, California, United States of America, **3** Department of Bioengineering and Therapeutic Sciences, University of California, San Francisco, California, United States of America, **4** Kavli Institute for Fundamental Neuroscience, University of California, San Francisco, California, United States of America, **5** Department of Pharmaceutical Chemistry, University of California, San Francisco, California, United States of America, **6** Institute for Neurodegenerative Diseases, University of California, San Francisco, California, United States of America, **7** Department of Epidemiology & Biostatistics, University of California, San Francisco, California, United States of America, **8** Investigator Program, Chan Zuckerberg Biohub SF, San Francisco, California, United States of America

* katherine.pollard@gladstone.ucsf.edu

## Abstract

The investigation of chromatin organization in single cells holds great promise for identifying causal relationships between genome structure and function. However, analysis of single-molecule data is hampered by extreme yet inherent heterogeneity, making it challenging to determine the contributions of individual chromatin fibers to bulk trends. To address this challenge, we propose ChromaFactor, a novel computational approach based on non-negative matrix factorization that deconvolves single-molecule chromatin organization datasets into their most salient primary components. ChromaFactor provides the ability to identify trends accounting for the maximum variance in the dataset while simultaneously describing the contribution of individual molecules to each component. Applying our approach to two single-molecule imaging datasets across different genomic scales, we find that these primary components demonstrate significant correlation with key functional phenotypes, including active transcription, enhancer-promoter distance, and genomic compartment. Also, we find that some bulk trends exist at the single-cell level, but only in a small fraction of cells, suggesting that critical changes in genome organization may be driven by specific rare subpopulations rather than occurring uniformly across all cells. ChromaFactor offers a robust tool for understanding the complex interplay between chromatin structure and function on individual DNA molecules, pinpointing which subpopulations drive functional changes and fostering new insights into cellular heterogeneity and its implications for bulk genomic phenomena.

## Author Summary

Emerging imaging technologies are generating high-resolution, single-molecule chromatin interaction data for individual loci, providing an opportunity to identify the chromatin

**Data availability statement:** All code used for the method, running experiments and creating figures is available on a GitHub repository at https://github.com/lgunsalus/ChromaFactor.

**Funding:** This work was supported by the National Institutes of Health 4D Nucleome Project (#U01HL157989 to KSP), the Chan Zuckerberg Initiative DAF (an advised fund of the Silicon Valley Community Foundation) (#2018-191905 to MJK), and the University of California San Francisco (Achievement Rewards for College Scientists Scholarship to LMG). The funders had no role in study design, data collection and analysis, decision to publish, or preparation of the manuscript. KSP and LMG received salary support from the National Institutes of Health. LMG received salary support from the University of California San Francisco. MJK received salary support from the Chan Zuckerberg Initiative.

**Competing interests:** The authors have declared that no competing interests exist.

structures that contribute to patterns seen in bulk genomics assays or that correlate with variability in gene regulation. However, chromatin conformation varies dramatically between cells, making it challenging to determine which patterns are important for transcriptional activity and which represent inherent heterogeneity. We present ChromaFactor, a new method to address this problem based on non-negative matrix factorization. ChromaFactor identifies the major interaction patterns across chromatin fibers as well as the individual molecules contributing to each pattern. We analyzed human and fly single-molecule imaging data, finding significant correlations between ChromaFactor's primary components and paired molecular phenotypes measured in the same cells, including active transcription, enhancer-promoter distance, and genomic compartment. Interaction patterns in primary components are also associated with bulk genomic measurements of transcription factor and structural protein binding. These findings demonstrate that ChromaFactor provides a robust framework for understanding the complex interplay between chromatin structure and function at the single-molecule level by treating single-cell chromatin measurements not as independent instances, but as snapshots of continuous, dynamic states.

## Introduction

Chromatin is intrinsically dynamic, and its behavior across time restricts and permits the precise regulatory landscape controlling gene expression [1,2]. Recent single-cell technologies such as single-cell Hi-C [3,4] and chromatin microscopy techniques [5–9] now offer unique insight into genome folding, allowing us to directly observe chromatin folding as well as functional readouts in individual cells to disentangle their mechanistic relationship.

Linking chromatin conformation to function in single cells presents several key challenges: 1) Single cell data is extremely sparse [10]. Current single-cell technology often yields incomplete information, such as missing values or misallocated genomic coordinates. 2) Single-cell measurements capture snapshots, whereas chromatin function may result from a dynamic behavior as it moves across time. Phenomena observed in bulk experiments, such as Hi-C, may be artifacts of averaging across cell populations and patterns seen in bulk may not exist in single cells [11]. 3) Capturing chromatin folding and phenotypic measurements like nascent transcription in the same cells has only recently become possible, but temporal offsets between folding and function could introduce uncertainty. 4) The bulk trends we observe in aggregate may be driven by a small fraction of cells, and identifying this subset amidst heterogeneous single-cell chromatin measurements presents a complex challenge. Connecting chromatin behavior to function in individual cells remains intractable given these technical barriers.

Several computational methods have been developed in response to emerging single-cell imaging and high-throughput sequencing techniques to measure chromatin conformation. These methods primarily focus on four key challenges: imputing and denoising sparse single-cell data [12–15]; clustering cells through topic modeling [16], random-walk methods [17], and recent deep learning approaches [12,18]; annotating chromatin structures in single cells, including A/B compartments [12,13], subcompartments [19], topologically associating domains (TADs) [12,13], and chromatin loops [20,21]; and linking chromatin structure to function, as demonstrated by Rajpurkar et al.'s convolutional neural network for predicting nascent transcription from chromatin folding [22] and Zhan et al.'s dimensionality reduction method [23]. However, relating patterns in single cells to those in bulk populations remains underexplored. Current methods do not yet connect the behavior of individual cells to populations of similar conformations that are transcriptionally on or off. We build on these works

to relate the behavior of individual cells to bulk trends, providing a framework to identify which subpopulations drive chromatin-function relationships observed in aggregate data. Our approach can be applied to a single locus without requiring genome-wide data for model training, making it directly applicable to emerging imaging technologies, though it can also be used to analyze single-cell Hi-C.

We introduce ChromaFactor, a non-negative matrix factorization (NMF) technique to decompose single-cell datasets into interpretable components and identify key subpopulations driving cellular phenotypes. Non-negative matrix factorization (NMF) offers an ideal application in the analysis of such complex data due to its inherent capacity to reduce high-dimensional data into a lower-dimensional, interpretable format [24]. NMF has a robust legacy in genomics as it allows for the deconvolution of composite signals into a set of additive components and can therefore discern patterns and structures in noisy, large-scale data. Notably, it has been used on bulk Hi-C data for TAD calling [25] and has found applications in other emergent single-cell modalities, including single-cell RNA-Seq [26] and spatial transcriptomics datasets [27]. By applying NMF to single-cell genome folding datasets, we can identify significant components or *'templates'* that account for the majority of cellular variation. Linking these templates to matched functional readouts describes how differences in cell populations correspond to differences in phenotypes. ChromaFactor deconvolves single-cell chromatin organization datasets into their most meaningful primary components, providing new insights into the interplay between chromatin structure and function. Here, we apply ChromaFactor to two single-cell imaging datasets and link templates to nascent transcription. This tool can also be applied to any set of ordered matrices in single cells, and may be set-up and run in minutes.

## Results

### NMF to decompose single-cell 3D genome conformation datasets

We were motivated to develop ChromaFactor by the disconnect between meaningful signal observed in bulk cell populations and the extreme heterogeneity of single-molecule examples. One such dataset, Mateo et al. 2019 [5], profiles local chromatin conformation at a single locus - the bithorax complex (BX-C) - in *Drosophila* embryos and additionally includes matched nascent transcription in the same cells ($n = 16,320$). To discover how cell populations vary, we often take the difference between the average contact maps under two conditions. We observed a pronounced boundary in cells within the 30 kb region actively transcribing the Abd-A gene, as compared to non-transcribing cells (Fig 1a). Statistical analysis confirms significant differences in contact patterns between transcribed and non-transcribed populations (Mann-Whitney U test with FDR correction, $p < 0.05$; S1 Fig), with the caveat that cells from the same sample may not be independent examples. However, these patterns are nearly impossible to discern in individual cells (Fig 1b). Given the heterogeneity and dynamic behavior of chromatin in single cells, identifying which cells contribute to overall trends is complex. Are these contact patterns visible at the single-cell level, or are they composite effects resulting from population-wide averaging? To bridge the gap between single-cells and bulk averages and identify which cells contribute most, we propose applying NMF to single-molecule chromatin conformation datasets with ChromaFactor.

In this approach, NMF decomposes a non-negative distance matrix into two lower-rank non-negative matrices, such that their product approximates the original matrix. ChromaFactor decomposes an $b$ by $b$ by $n$ count matrix, where $b$ is the number of genomic loci, or bins, profiled and $n$ is the number of cells, into an $b$ by $b$ by $k$ component matrix W, where $k$ is a specified number of components, and a $k$ by $n$ weight matrix, H (Fig 1c). The matrix W represents the basis vectors,

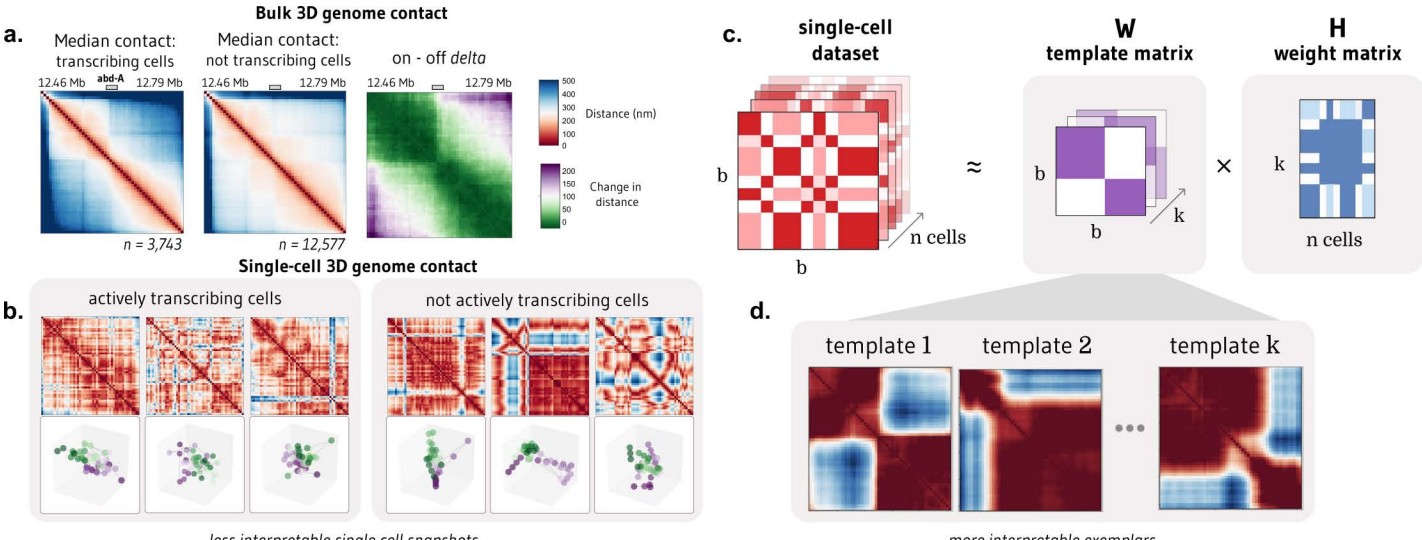

**Fig 1. NMF provides interpretable decomposition of single-molecule chromatin conformation datasets.** a. Matrices representing the median all-by-all Euclidean difference (nm) between genomic loci in single molecules at a 30 kb locus in *Drosophila melanogaster* actively transcribing (left) and not transcribing (middle) the Abd-A gene from Mateo et al.[5] ($n = 16,320$ molecules). The rightmost panel shows the difference in distance matrices, indicating two domains with elevated local interactions and reduced distal interactions in populations transcribing Abd-A. **b.** Bulk trends in contact change are challenging to observe in single cells actively transcribing (left) and not transcribing (right) Abd-A. **c.** Non-negative matrix factorization (NMF) decomposes a dataset of single-cell distance matrices into a *template* matrix with interpretable chromatin domain boundaries and a *contribution* matrix describing the weight of each template to each cell. **d.** Three templates produced when NMF is applied to distance matrices at the Abd-A locus.

which we call *templates*, as they resemble patterns observed across the cell population. The method accepts both distance matrices from single-molecule imaging experiments and contact matrices from single-cell Hi-C. We introduce a selection procedure for the number of components ($k$), called *KSelector* that enables users to balance reconstruction error, component stability, component redundancy, computational efficiency, and correlation with transcription, when matched data is available (**Methods and** S2 Fig). The matrix H represents the weight matrix, signifying contributions of cells to components such that the data for each molecule is approximated as the weighted average of the components plus noise. To estimate W and H, matrices are randomly initialized and updated to minimize the reconstruction error between their product and the single-cell dataset.

## Relating single cells to bulk trends with ChromaFactor

When ChromaFactor is applied to the Mateo et al. [5] dataset with twenty components, we find that several templates resemble chromatin boundaries (**Methods**, Figs 1d and S3). To quantitatively validate these boundaries, we computed insulation scores across each template and examined their correspondence with known architectural proteins (S4 Fig **and Methods**). Analysis of ChIP-seq data for boundary-associated proteins (CTCF, cohesin subunit Rad21, and insulator CP190) revealed strong colocalization with template boundaries in templates 0, 1 and 5. Most boundaries show concurrent binding of all three proteins, though some locations exhibit CTCF and CP190 binding without Rad21. This pattern suggests that different cell subpopulations may rely on different combinations of these architectural proteins to establish alternative boundary configurations, potentially explaining the distinct boundary patterns observed across templates.

To visualize the relationships within the single-cell dataset, we apply UMAP on the weight matrix, H (Fig 2a), and label cells by the component with the largest weight. Investigating

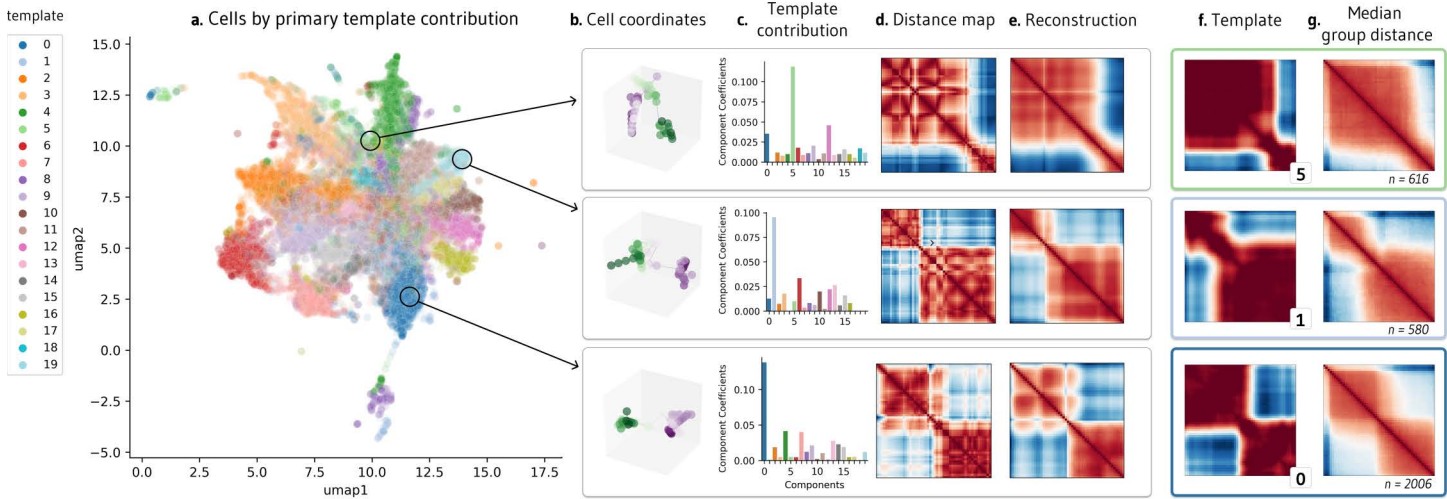

**Fig 2. Visualization of NMF outputs and their relationship to single-cell behavior.** a. UMAP visualization of contribution matrix, colored by the template with the predominant contribution in each cell. **b.** Depiction of cell coordinates from selected individual cells. **c.** Component contributions for each cell, emphasizing high weight for templates 5, 1, and 0. **d.** Distance matrices corresponding to each cell. **e.** Denoised reconstructions of the distance matrices, created by multiplying the template contribution of a cell shown in (c) by the template matrix. **f.** NMF templates 5, 1, and 0, which had the highest weight contributions for the three individual cells in (c). **g.** Median contact distance across all cells in the dataset with the highest weight contributions to templates 5, 1, and 0, respectively.

individual examples, we find that single cells can resemble these template patterns. To illustrate, we show the 3D coordinates and distance maps of three cells, along with their component contributions from weight matrix H (Fig 2b-d). Since every cell is an additive combination of the templates, we can multiply each cell's weights by the component matrix to reconstruct single-cell examples, which are less noisy than the original cell measurements (Fig 2e). Notably, these cells closely resemble the component with the highest contribution, indicating that templates can be representative of cell subpopulations (Fig 2f). Indeed, considering all cells in the same group, we find that median contact resembles the closest template (Fig 2g)

## Templates are significantly correlated with active transcription

Templates may capture subsets of cells and cellular patterns. Which components, if any, correspond to biological phenotypes? Clustering the transcribing and non-transcribing cells based on their component weights suggests that these components capture biologically meaningful chromatin states (S5 Fig). To investigate if templates are correlated with downstream biological function, we train a random forest model to predict nascent transcription of nearby genes from the weight matrix H alone (**Methods and** Fig 3a). In the Mateo et al. dataset, three genes were profiled in the same cells that were imaged, producing matched chromatin organization and transcription data[5]. Predictive performance would indicate that the components capture salient information about transcriptional state and may serve as a proxy for the raw input distance matrices themselves. We designed this analysis specifically to identify which templates are biologically relevant to transcription, rather than to optimize predictive performance. By using balanced datasets with equal numbers of transcribing and non-transcribing cells, we can easily assess that performance exceeds random chance (50%) and represents biological signal.

Different random forest models were separately trained to predict transcription in the 17 measured gene isoforms. We find that the weight matrix can modestly predict transcription

across several genes, including Abd-a, Ubx, and Abd-b on balanced datasets of transcribing and non-transcribing cells (Fig 3b). Indeed, the performance of the random forest parallels the performance of a random forest trained directly on the distance matrices, achieving an accuracy of 67.4% and 65.3%, respectively. Examining the feature importance of the Abd-a trained model, we find that components 0, 1, 5, and 14 are particularly influential for the model's prediction (Fig 3c). Permutation testing confirms this predictive performance is significant for Abd-A while remaining equivalent to random guessing for unrelated genes (S6 Fig).

We can alternatively address which components are preferentially upweighted by transcribed cells by evaluating component weights separately for transcribing and non-transcribing cells. Eighteen of the twenty components showed significantly different weights between Abd-A transcribing and non-transcribing cells after controlling for multiple testing (FDR < 0.1), with most components showing very strong significance (FDR < 0.001). Components 0, 1, 5, and 14, which were also identified as salient by the random forest model, showed particularly strong differences (FDR = 0.000), Fig 3d). Visually, these components show a separation of chromatin into two distinct compartments across three separate points across the locus (components 0, 1,

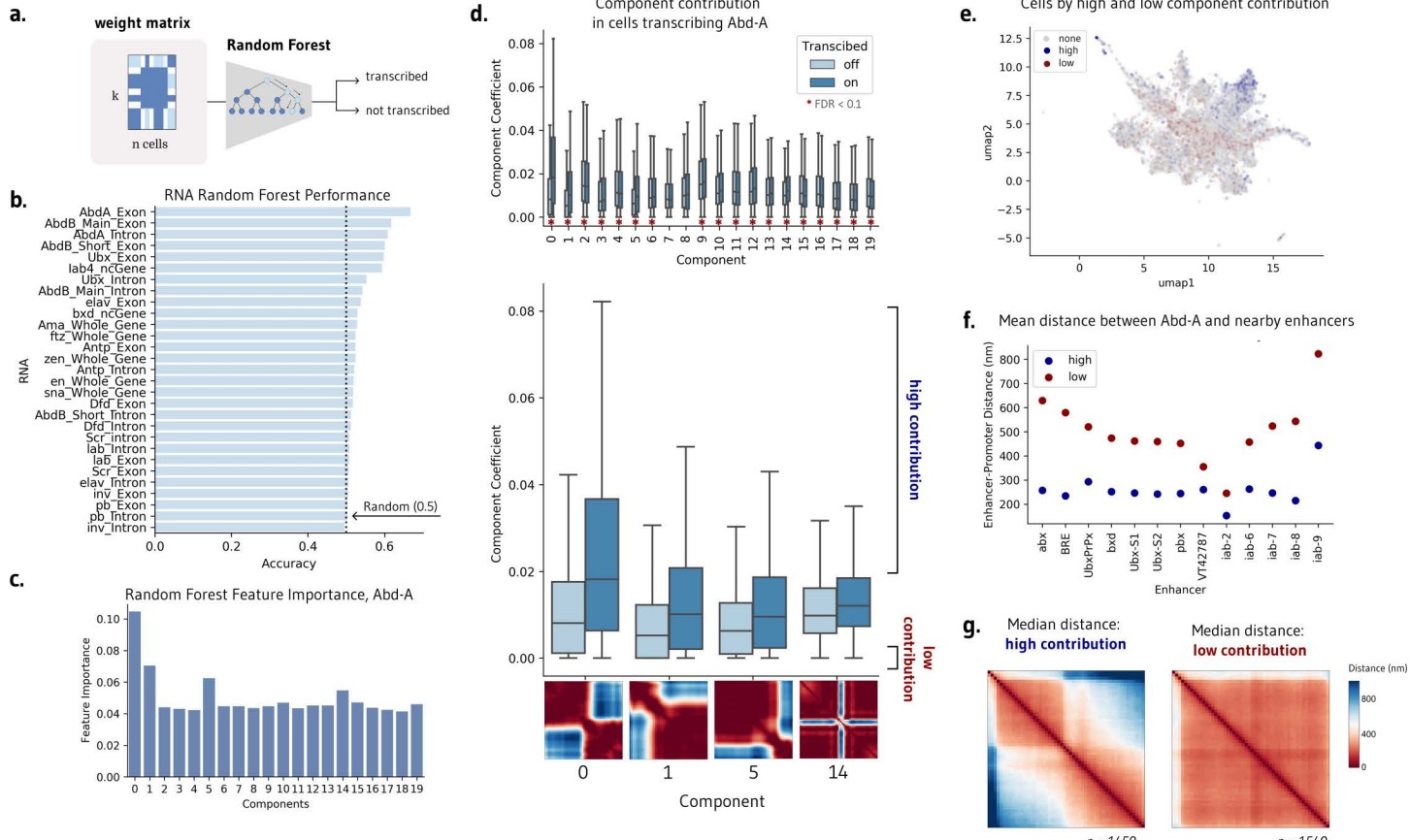

**Fig 3. NMF templates are significantly correlated with transcription.** a. Application of random forest models to predict cell transcription from the contribution matrix alone. **b.** A random forest can modestly predict transcription in abd-A, Abd-B, and Ubx, demonstrating that the components capture salient information for transcription. **c.** Random forest feature importance highlights templates 0, 1, and 5 as most important for predicting transcription. **d.** Several components, including 0, 1, 5, and 14, have significantly different component contribution weights in transcribing and non-transcribing cells (FDR < 0.001, Benjamini-Hochberg correction). **e.** UMAP visualization of component contribution matrix, colored by cells with a high contribution of components 0, 1, 5, and 14 (blue) and a low contribution of these components (red). **f.** Mean distance between abd-A and nearby enhancers at the same locus across the subset of cells with high and low component contributions. **g.** Median contact of cells with high and low component contributions, encompassing the subset of cells which may be responsible for changes in contact observed 0 in bulk.

5), as well as a sharp decrease in contact at the center of the locus (component 14). Significant templates suggest how contact differs upon active transcription to favor stricter subcompartmentalization within the genomic locus.

## Subpopulations of transcribing cells drive contact patterns observed in aggregate

Our aim is to understand not just how contact differs, but which subpopulations of cells drive the changes we observe in bulk (Fig 1a). We consider transcribing cells with the top 50% of weights in components 0, 1, 5, and 14, which we call 'high contribution' cells, and contrast them with non-transcribing cells in the bottom 50% of component contributions ('low contribution' cells). These cells make up only 12.7% of the total cell population, but their component weights are the most predictive of transcriptional state. These high and low contribution cells occupy different areas of the UMAP plot seen previously– low contribution cells are more likely to be mixes of components and high contribution cells are more likely to favor one component, suggesting that biologically consequential cells resemble templates (Fig 3e).

These subpopulations of cells differ not just in chromatin conformation and transcriptional state, but also in their local behavior of regulatory elements. The distance between Abd-A and all proximate enhancers at the same locus is notably smaller in high contribution cells as compared to low contribution cells (Fig 3f). Enhancers are closer to the gene promoter in the subset of active cells identified by ChromaFactor. Moreover, examining the median contact of high and low contribution cells, we observe contact patterns far more pronounced than those observed in bulk (Fig 3g). Cells with high component weights possess stronger boundary separation as well as a stripe of contact centered at the location of Abd-A, which is absent in low-contribution, transcriptionally-off cells. In this case, the cell population identified by ChromaFactor exhibits a more potent and unified profile of compartmentalized chromatin driving smaller enhancer-promoter distances when compared with all transcribing cells.

In sum, template analysis at this locus paints a holistic portrait of higher genomic sequestration between loci, reducing the distance between enhancers and promoters, thereby increasing the likelihood of transcription. This trend, although suggested at the level of the bulk population, is strongly driven by a small subpopulation of single cells. The remaining population is extremely heterogeneous across transcribing and non-transcribing cells such that their contact effectively cancels out.

## Application of ChromaFactor to holocarboxylase synthetase (HLCS) locus highlights local and compartment-level chromatin shifts upon transcription

To demonstrate the efficacy of ChromaFactor across genomic scales, we next apply NMF to a 10 Mb locus in human IMR90 cells with 40 kb resolution[6]. We profile a population of 7,590 cells derived from genome-wide profiling of chromatin conformation and nascent transcription with microscopy. The median genomic distance between cells actively transcribing and not transcribing the HLCS gene reveals no visually discernible change in contact (Fig 4a). However, after subtracting one contact matrix from the other to examine the difference in contact between populations, we observe a weakened boundary directly upstream of the HLCS locus.

We decompose the single-cell imaging dataset into twenty components with ChromaFactor and identify components with significantly different weights in transcribing cells (Figs 4b and S7). Of the twenty components, five showed significant differences between transcribing and non-transcribing cells after controlling for multiple testing (FDR < 0.1), exhibiting a diverse range in contact differences across templates. Component weights are elevated in transcribing

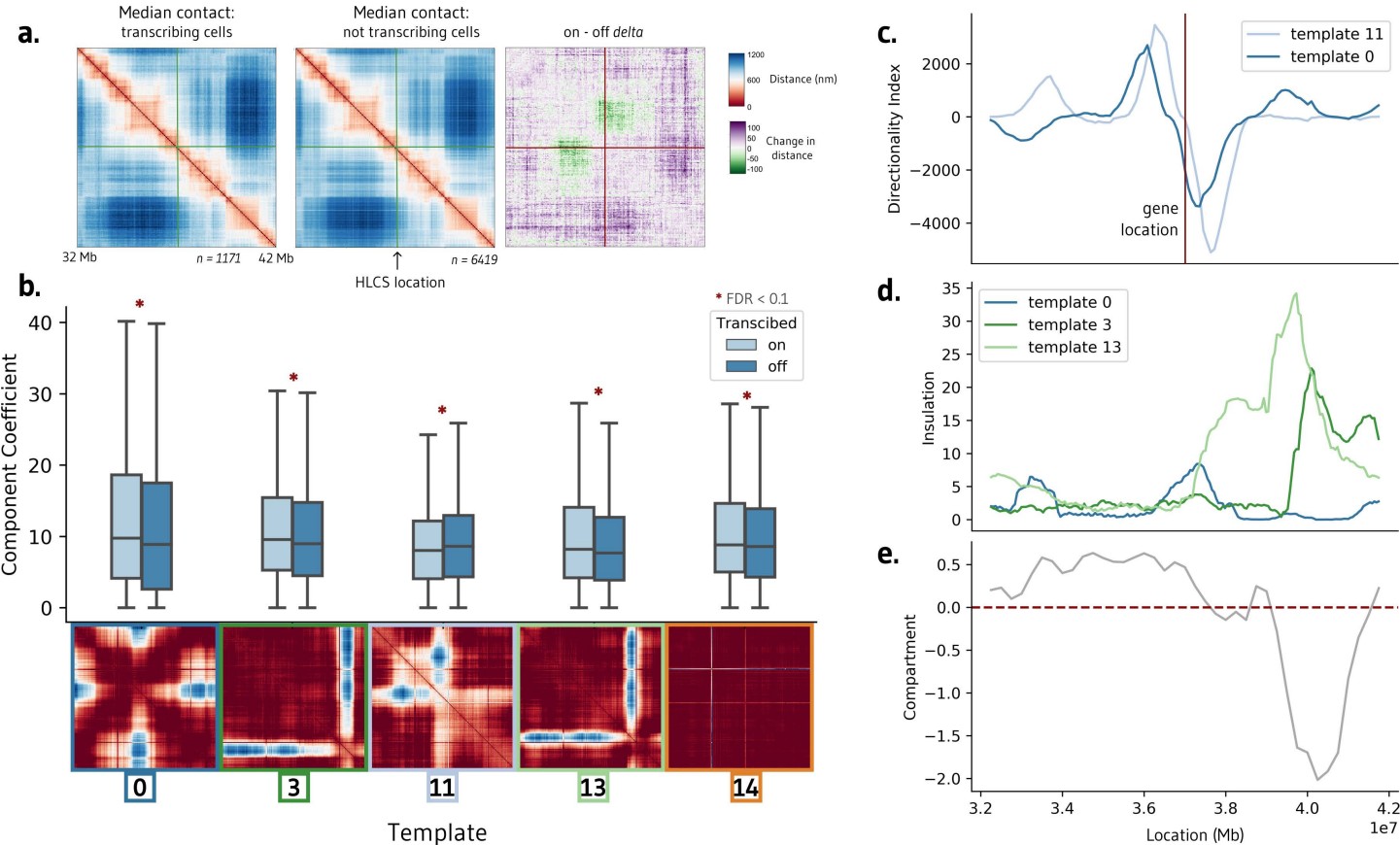

**Fig 4. Interpretable templates at the HCLS locus in IMR-90 cells.** a. Average chromatin contacts in cells actively transcribing (left) and non-transcribing (middle) the HCLS gene within the surrounding 10 Mb region. The right panel highlights the contrast in contact patterns, emphasizing a stronger boundary in actively transcribing cells. **b.** Templates generated using Non-Negative Matrix Factorization (NMF) on this cell population. Among the 20 components, five exhibit significant differences between cells transcribing and not transcribing the HCLS gene (FDR < 0.1, Benjamini-Hochberg correction). **c.** Directionality index of templates 0 and 11 correspond with the location of the transcribed HLCS gene. d-e. Insulation scores of templates 3 and 4 align with a shift in compartment in IMR-90 cells.

cells in all components but component 11, where a decrease in contact is observed at the precise location of the boundary loss observed in bulk. Curiously, component 14 highlights a sharp change in contact at two particular genomic loci, one of which is the location of the HLCS gene itself. The method has no knowledge of transcriptional state nor gene location, indicating that change in contact at this locus may nonetheless contribute to a significant amount of variation across the cell population.

We find that two significant components, 0 and 11, correspond to a steep shift in the directionality index, a measure of contact frequency bias towards either upstream sequence or downstream sequence, at the location of the HLCS locus (Fig 4c). Directionality index indicates regions of highly biased interaction frequency, which commonly occurs at domain boundaries [28]. Previous studies have shown that the loss or gain of these boundaries can shift the interaction frequency between enhancers and their target genes, impacting transcription[2]. Remaining components 3 and 13 display a sharp increase in insulation at the location of a compartment switch from A to B in IMR90 cells (Fig 4d and 4e). In sum, template analysis at this locus suggests that the change of transcription correlates with a reorganization of chromatin directly at the site of active transcription, at boundary shifts directly upstream of the locus, and within broader compartments downstream of the locus.

We additionally investigated a 10Mb locus containing multiple genes to demonstrate how template analysis reveals patterns masked in population-level analysis[6] (S8 Fig). We examined two nearby genes, BACE2 and SON, which show similar changes in their bulk difference contact maps. Our analysis revealed that BACE2 transcription appears to require a minimal contribution from one component, while SON's activity is driven by a small subset of cells with exceptionally high weights for a different component. These distinct regulatory strategies would be masked in traditional bulk analysis, demonstrating how template decomposition can reveal mechanistic insights into gene regulation.

## Discussion

This study has demonstrated the utility of ChromaFactor, a novel application leveraging Non-negative Matrix Factorization (NMF) for dissecting single-cell chromatin conformation datasets. Our method uncovers nuanced layers of genome conformation dynamics and their correlation with transcriptional states, which would otherwise be obscured in bulk analyses. Correlations between template patterns and active transcription suggest that these templates are not merely reflections of cellular heterogeneity, but could be mechanistically linked to transcriptional regulation. Our application of ChromaFactor to the Mateo et al. [5] and Su et al. [6] datasets leads us to two intriguing biological interpretations: 1) bulk behavior can sometimes be observed in individual cells, and 2) only a small minority of cells in the population drive population-wide signal. This reasoning is not possible at the bulk level, where trends may be artifacts of averaging, nor at the single-cell level, where it is unknown which snapshots capture relevant signal.

The application of ChromaFactor to a wider array of cell types, genomic phenomena, and sc-HiC will help us to understand if these conclusions hold across biological contexts. ChromaFactor's ability to isolate functional portions of single-cell chromatin datasets could clarify the structural dynamics underpinning cell-type-specific gene regulation and the development of cellular heterogeneity. Additionally, integrating ChromaFactor with multi-omics approaches, such as single-cell RNA-Seq, ATAC-Seq, or CUT&Tag, could help resolve the interplay between chromatin structure, epigenetic modifications, and gene expression. ChromaFactor's application to pathological states or CTCF degradation experiments presents another exciting area of exploration.

While the application of ChromaFactor is promising, it is important to note its current limitations. The approach also relies heavily on the quality of the single-cell chromatin datasets and the resolution at which they are produced. The inherent noise and technical artifacts present in these datasets can influence the deconvolution process and the interpretation of the resulting components. Extremely noisy datasets with consistent dropout locations will produce templates capturing dropout. Further improvements in single-cell chromatin imaging and sequencing technologies will likely enhance the accuracy and interpretability of ChromaFactor's outputs.

Finally, a deeper investigation of the biological interpretation of components is warranted. While we have shown that these components correlate with transcription and other genomic features, the exact mechanisms through which these templates influence cellular behaviors remain largely unknown. Additional studies are needed to mechanistically link these structural templates with specific functional outputs and to explore their potential role in modulating regulatory response. In sum, this study introduces ChromaFactor as a promising tool for decoding single-cell chromatin conformation data. It provides a more granular view of the dynamic nature of genome architecture and its role in gene regulation, thereby broadening our perspective on the intricate interplay of genome organization and function.

Looking ahead, ChromaFactor suggests a shift in how we conceptualize single-cell chromatin measurements with computational approaches. Current methods view data from single-cell Hi-C and imaging-based experiments as independent examples. However, they are more realistically understood as samples from a continuous and dynamic structure whose functional states are probabilistically constrained by its three-dimensional organization. ChromaFactor's ability to identify cell subpopulations that drive specific patterns represents an initial step toward viewing chromatin organization as a spectrum of plausible conformational states that influence cellular function. Rather than calling structures on sparse individual snapshots, we can begin to understand them in the context of sets of related conformational states. Just as proteins sample different conformational states that collectively determine their binding affinities and catalytic activities, chromatin organization may be better understood as an ensemble of related states that together influence regulatory potential. This perspective could guide the development of computational methods that approximate continuous chromatin dynamics in single cells - an important step toward the ultimate goal of directly capturing how the 3D genome moves across time.

## Methods

### Datasets and processing

**Mateo et al. dataset.** The Mateo et al. microscopy dataset contains 3D genomic coordinates and transcriptional activity for single molecules spanning the Drosophila Bithorax complex (BX-C) locus [5], which can be found at the following repository: https://zenodo.org/records/4741214. We employed the data preprocessing procedure from Rajpurkar et al. [22] to handle missing values, which can be found at the following repository: https://github.com/aparna-arr/DeepLearningChromatinStructure/tree/master/DataPreprocessing.

Cells with over 80% of coordinates missing were excluded from our analysis. For the remaining cells, missing coordinates were imputed by linear interpolation between adjacent loci using the scipy.interpolate.interp1d function [29]. Maps were normalized by dividing by the maximum distance observed as followed prior to NMF. The Mateo dataset originally consists of 19,103 cells, of which 16,320 were used for analysis after quality control filtering.

**Su. et al. dataset.** The Su et al. dataset [6] comprises genome-wide chromatin folding data from single molecules in human IMR90 fibroblasts imaged using DNA FISH. Data can be downloaded from the following repository: https://zenodo.org/record/3928890. In particular, we analyze paired coordinate and transcription data in 'genomic-scale-with transcription and nuclear bodies.tsv'. Custom Python code was written to extract specific genomic regions from the raw dataset, transform coordinate data into distance matrices, and identify cells transcribing the HLCS (ENSG00000159267) gene for downstream analysis, and can be found in the provided repo with an example of processing. Maps were considered with a 40k resolution. Cells with more than 25% missing coordinates within the 10 Mb HLCS genomic locus were excluded. Any remaining missing values were imputed by linear interpolation using numpy.interp [30].

In both cases, ChromaFactor was applied to analyze chromatin conformation at individual genomic loci (10Mb or smaller) rather than genome-wide to focus investigation of local chromatin organization and transcriptional activity.

### Baseline significance testing

To statistically validate differences in contact patterns between transcribed and non-transcribed cells, we performed Mann-Whitney U tests at each position in the contact matrix. For each position (i,j), we compared the distribution of contact values from transcribed cells versus non-transcribed cells. P-values were corrected for multiple testing using the Benjamini-Hochberg

procedure to control the false discovery rate. Positions with FDR < 0.05 were considered significant. This analysis confirmed statistically significant differences in contact patterns between populations, particularly in regions highlighted in Fig 1a (S1 Fig).

## Non-negative matrix factorization (NMF) and ChromaFactor application

We applied non-negative matrix factorization (NMF) to decompose single-cell chromatin conformation data using the ChannelReducer wrapper in the Lucid NMF library [31] built on top of the scikit-learn implementation [32]. While standard NMF decomposes a non-negative matrix V into two lower-rank matrices W and H such that $V \approx WH$, our input data consists of n cells, each containing a $b \times b$ distance matrix, forming a $b \times b \times n$ tensor X. The ChannelReducer wrapper handles this higher dimensionality by transforming the input tensor $X \in R^{(b \times b \times n)}$ into a 2D matrix $V \in R^{(b^2 \times n)}$ by flattening each distance matrix into a vector of length $b^2$. This flattened matrix V is then decomposed using standard NMF into matrices $W \in R^{(b^2 \times k)}$ and $H \in R^{(k \times n)}$, where k is the number of components.

After decomposition, the components matrix W is reshaped back into a $b \times b \times k$ tensor, where each k slice represents a template pattern across the genomic locus, while the weight matrix H remains $k \times n$, with each column representing the contribution of each template to a given cell. This reshaping approach preserves all spatial relationships between genomic loci during flattening and reconstruction, allowing the same weight matrix H to reconstruct the distance matrices of all cells while maintaining non-negativity constraints throughout.

We set the number of components to k=20 to balance interpretability of templates and reconstruction error (S2 Fig). Default scikit-learn NMF parameters were used: NNDSVD initialization, a coordinate descent solver, Frobenius loss, tolerance of 1e-4, maximum 200 iterations, and an element-wise L2 regularization penalty. The NNDSVD (Non-Negative Double Singular Value Decomposition) initialization was chosen as it provides a deterministic, sparse initialization based on the SVD of the input data, making it particularly suitable for sparse biological datasets. The coordinate descent solver was selected for its efficiency with the Frobenius norm objective function, which measures the element-wise reconstruction error between the original and factorized matrices.

Reconstruction of individual cell matrices can be achieved by computing $X_i \approx \Sigma W_k H_{ki}$, where $X_i$ is the distance matrix for cell i, $W_k$ represents template k, and $H_{ki}$ is the weight of template k in cell i. Additional code is provided to process both datasets and plot 2D distance matrices and 3D coordinates. The complete implementation, including data preprocessing and visualization code, is available in our GitHub repository.

## Selection of number of components

To systematically determine the optimal number of components (k), we developed a comprehensive evaluation framework, *KSelector*, that considers multiple metrics: reconstruction error, component stability, component redundancy, computational efficiency, and biological significance. The reconstruction error measures how well the NMF approximation matches the original data. Component stability is assessed by running multiple NMF initializations and measuring the correlation between resulting components. Component redundancy evaluates whether additional components capture unique patterns or become redundant. When transcription labels are available, we also evaluate how well the components predict transcriptional state using a random forest classifier.

For a given range of k values, each metric is calculated and normalized. Reconstruction error is calculated as the Frobenius norm between the original and reconstructed matrices. Stability is measured as the average correlation between components across multiple random

initializations. Component redundancy is computed as the mean absolute correlation between all pairs of components within a given k. When applicable, transcription prediction accuracy is assessed through cross-validated random forest classification.

The framework provides quantitative metrics, plots these results across a range of k choices, and suggests a proposed k by averaging across these metrics. While an elbow in reconstruction error occurs around k=20, suggesting diminishing returns for additional components, the final selection of k balances multiple criteria including stability, interpretability, and biological significance. Results across all metrics are stored to enable reproducible analysis and comparison across datasets.

### Random forest

We trained random forest classifiers using RandomForestClassifier in scikit-learn [32] to predict transcriptional activity from NMF component weights. Models were trained separately for each gene with binary on/off transcription labels. Balanced datasets were created for each gene with equal transcribing and non-transcribing cells. Data was split 70/30 into train and test sets. All other random forest parameters were left as scikit-learn defaults. Balanced datasets with equal numbers of transcribing and non-transcribing cells were used to ensure 0.5 represents true random performance. Each permutation maintained the same proportion of transcribing to non-transcribing cells while randomly reassigning labels. Random forest parameters for shuffled permutations were identical to those used in the main analysis.

### Statistical analysis

Differences in contact patterns and NMF component weights between transcribing and non-transcribing cells were evaluated using the non-parametric two-sided Mann-Whitney U statistical test. To account for multiple comparisons when testing *k* components, we applied Benjamini-Hochberg false discovery rate (FDR) correction. Components with FDR < 0.1 were considered significant. For the Abd-A locus analysis, eighteen components showed significant differences after FDR correction (FDR < 0.05 for most components). For the HLCS locus analysis, five components showed significant differences after FDR correction (FDR = 0.021–0.091).

### Component weight analysis

To visualize patterns in the H matrix (S5 Fig), we selected a balanced subset of 1000 cells (500 transcribing, 500 non-transcribing) with the highest component contributions across any single component. Hierarchical clustering was performed on both rows (components) and columns (cells) using seaborn's clustermap function to reveal patterns in component weights associated with transcriptional state. UMAP dimensionality reduction for visualization used the scikit-learn implementation with n_neighbors=5 and remaining default parameters.

### Boundary analysis and protein binding

To identify and validate chromatin boundaries in templates, we computed insulation scores using a sliding window approach [33,34]. For each position along the matrix diagonal, we calculated the average contact frequency within each bin. Local maxima in this insulation profile indicate potential boundary locations. We analyzed ChIP-seq data for three boundary-associated proteins: CTCF, Rad21, and CP190, which were measured in the *Mateo et al.* study [5]. For each probe bin location, we considered the strongest peak intensity for each protein. Binding sites were compared to template boundaries by identifying regions where insulation scores exceeded 0.05 and examining protein binding patterns at these locations.

### Genomic features annotations

Gene annotations and enhancer locations were derived from the original publications for each dataset. Enhancer and gene locations were provided from Mateo et al [5]. Compartment annotations were used from Rao et al. [35] (4DNFIHM89EGL). Directionality index and insulation tracks were produced from scoring code provided in Gunsalus et al [36].

## Supporting information

**S1 Fig.  Statistical comparison of contact patterns between transcribed and non-transcribed cells.** A. Heatmap showing -log10(p-values) from Mann-Whitney U tests comparing contact values at each matrix position between transcribed and non-transcribed cells shown in Fig 1, with Benjamini-Hochberg correction for multiple testing. Darker colors indicate more significant differences. **B.** Overlay of significant positions (FDR < 0.05, outlined in white) on the median difference map from Fig 1a.
(TIFF)

**S2 Fig.  Systematic evaluation of optimal number of components (k) A.** Reconstruction error (Frobenius norm) between original and NMF-approximated matrices across different k choices. The red dashed line indicates the elbow point where additional components yield diminishing returns. **B.** Computational time (in second) required for NMF decomposition. **C.** Component stability measured as the average correlation between components across multiple random initializations, with higher values indicating more stable templates. **D.** Component redundancy calculated as the mean absolute correlation between component pairs. Lower values suggest components capture distinct patterns. **E.** Variance explained showing the fraction of total variance in the data captured by k components. **F.** Transcription prediction accuracy using random forest classification with component weights as features, demonstrating the biological relevance of the decomposition.
(TIFF)

**S3 Fig.  Components at BX-C locus.** All 20 components generated by applying NMF across the cells at this locus from the Mateo et al. dataset.
(TIFF)

**S4 Fig.  Validation of chromatin boundaries in templates with architectural proteins.** Insulation scores computed across templates 0, 1, and 5 overlaid with ChIP-seq binding locations for CTCF (red), Rad21 (blue), and CP190 (green). Vertical lines indicate protein binding sites at locations where template insulation score exceeds 0.05. Multiple lines at the same position indicate concurrent binding of different proteins.
(TIFF)

**S5 Fig.  Heatmap visualization of the H matrix component weights.** We selected a balanced subset of 500 transcribing and 500 non-transcribing cells with the highest component contributions. Rows (components) and columns (cells) were hierarchically clustered. Cells are labeled by transcriptional state (transcribing/non-transcribing), revealing patterns of component weights that correlate with transcriptional activity.
(TIFF)

**S6 Fig.  Permutation testing validates template specificity for Abd-A transcription A.** Random forest prediction accuracy for Abd-A transcription using original data (0.671) compared to 10 permutations with randomly shuffled transcription labels (mean $0.498 \pm 0.003$, $p < 2.2 \times 10^{-16}$). The significant drop to random chance performance demonstrates that the original

predictive signal is not arising by chance. **B.** As a negative control, random forest prediction accuracy for inv_Intron transcription using original data (0.511) compared to 10 permutations with randomly shuffled labels (mean 0.500 ± 0.002). Both original and permuted performances are effectively equivalent to random guessing, confirming that templates show specificity for relevant genomic loci.
(TIFF)

**S7 Fig. Components at HLCS locus.** All 20 components generated by applying NMF across the cells at this locus from the Su et al. dataset.
(TIFF)

**S8 Fig. Templates reveal distinct regulatory strategies masked in population-level analysis.** A. Average contact map across all cells in a 10Mb region of chromosome 21 containing SON, RUNX1, SIM2, ETS2, and BACE2 genes. **B-F.** Differential contact maps (transcribing minus non-transcribing cells) across a 10Mb locus. SON (B) and (C) BACE2 (C) demonstrate similar patterns of contact changes at the population level. **G.** Mean differences in component weights between transcribing and non-transcribing cells for all five genes across 20 components. Black boxes indicate significant differences (FDR < 0.05, Benjamini-Hochberg correction), revealing that SON and BACE2 utilize different sets of significant components despite similar bulk contact changes. **H.** Templates with significant differential weights for SON and BACE2, illustrating distinct structural patterns associated with transcription of each gene. **I.** Distribution of component 19 weights in SON-transcribing and non-transcribing cells. A small subset of non-transcribing cells shows very low weights (left arrow), while a distinct population of transcribing cells exhibits exceptionally high weights (right arrow). **J.** Distribution of component 17 weights for BACE2, revealing a population of non-transcribing cells with zero weight (arrow) while all transcribing cells maintain positive weights, indicating this component may represent a necessary chromatin configuration for BACE2 transcription.
(TIFF)

## Acknowledgments

We thank Chris Olah for the original idea to apply NMF to single-cell 3D genome datasets and Kangway Chuang for project guidance, including the idea to train a random forest to test component significance. We additionally thank Archit Verma, Amanda Everitt, Katie Gjoni, Shuzhen Kuang, and other members of the Pollard and Keiser labs for helpful discussion and manuscript feedback.

## Author contributions

**Conceptualization:** Laura M. Gunsalus, Michael J. Keiser, Katherine S. Pollard.

**Data curation:** Laura M. Gunsalus.

**Formal analysis:** Laura M. Gunsalus.

**Funding acquisition:** Michael J. Keiser, Katherine S. Pollard.

**Investigation:** Laura M. Gunsalus.

**Methodology:** Laura M. Gunsalus, Michael J. Keiser, Katherine S. Pollard.

**Project administration:** Katherine S. Pollard.

**Resources:** Michael J. Keiser, Katherine S. Pollard.

**Software:** Laura M. Gunsalus.

**Supervision:** Michael J. Keiser, Katherine S. Pollard.

**Validation:** Laura M. Gunsalus.

**Visualization:** Laura M. Gunsalus.

**Writing – original draft:** Laura M. Gunsalus.

**Writing – review & editing:** Michael J. Keiser, Katherine S. Pollard.

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
