## [Decision Letter · Decision Letter 0]

25 Jun 2024

Dear Dr. Pollard,

Thank you very much for submitting your manuscript "ChromaFactor: deconvolution of single-molecule chromatin organization with non-negative matrix factorization" for consideration at PLOS Computational Biology.

As with all papers reviewed by the journal, your manuscript was reviewed by members of the editorial board and by several independent reviewers. In light of the reviews (below this email), we would like to invite the resubmission of a significantly-revised version that takes into account the reviewers' comments.

We cannot make any decision about publication until we have seen the revised manuscript and your response to the reviewers' comments. Your revised manuscript is also likely to be sent to reviewers for further evaluation.

Sincerely,

Jie Liu

Academic Editor

PLOS Computational Biology

Jian Ma

Section Editor

PLOS Computational Biology

Reviewer's Responses to Questions

**Comments to the Authors:**

Reviewer #1: ChromaFactor describes a non-negative matrix factorization method on single-cell chromatin contact matrices that deconvolves contact matrices to primary components of contact variations and their cellular contribution. Such components are then associated with cellular transcriptional profiles. While the general idea of associating patterns of chromatin contacts with transcription is interesting, several major issues need to be addressed:

1. The authors demonstrate that their method provides a clearer picture than bulk-level analysis through visualization, which is not surprising given that the bulk-level analysis ignores existing single-cell variations. A more acceptable baseline would be to derive transcription-associated contact patterns between the transcribed vs non-transcribed cell populations and compare that with the result of ChromaFactor.

2. Other matrix decomposition methods exist on single-cell Hi-C datasets, such as Higashi (Zhang et al. 2022) or Topic modeling (Kim et al. 2020). What is the methodological advantage of ChromaFactor compared to other matrix factorization methods? It would be interesting to compare ChromFactor’s reconstruction error and components with those methods. 

3. Is the method applied to a single chromosome or across all chromosomes?

4. Besides visualization through UMAPs colored by the predominant component (Figures 2a and 3e), does matrix H capture known biological variations among cells?

5. Although the matrix decomposition step only requires contact matrices, the method's primary application seems to rely on transcriptional profiles from corresponding cells. Some clarification and discussion of the limitation is needed.

6. I cannot find multiple hypothesis corrections of all the reported p-values.

7. On the second paragraph of Page 3, it is stated that “we find that several templates resemble chromatin boundaries”. It is unclear to me what boundaries they are or how this claim is supported.

8. Regarding transcription prediction using random forest models, it is unclear what the dotted line in Figure 3b is and what a random baseline accuracy is. Also, the current performance might be limited by the default parameters. Would the performance improve with parameter tuning with nested cross-validation? Would a regression model result in better predictive power?

9. Associating transcription with contact patterns is the most interesting part, though such analysis's advantages or biological insights are not adequately explained or validated. Are these identified components representing general cell state changes or specific to the expression of Abd-A and HLCS? What does “a steep shift in the directionality index” inform regarding transcriptional regulation (Figure 4c)?

Reviewer #2: In this manuscript, the authors propose an NMF-based framework, ChromaFactor, to decompose single-cell datasets into different components and identify important cells influencing cellular phenotypes. By applying ChromaFactor to two single-cell genome folding datasets, it reveals patterns and structures, linking these to functional outputs like nascent transcription. The proposed framework is easy to understand, and the GitHub link is provided. However, I have the following concerns about the manuscript.

1. There is inconsistency in the notation used for cells between different sections of the manuscript. For example, 'n' is used for cells in the first paragraph of the Results section, while 'm' is used in Figure 1. Additionally, the number of cells in the Mateo dataset is unclear. it is 19,103 or 16,320? This discrepancy should be clarified to avoid confusion.

2. The criteria for determining the number of components (k) are not well explained. From Supplementary Figure 1, it is difficult to discern why k=20 was chosen. It would be better to show the performance of the framework with different numbers of components to provide a clearer justification for selecting k=20.

3. The classification accuracy shown in Figure 3b is close to random, making it challenging to identify which feature is more important. Additional baseline methods are needed to evaluate the framework's performance.

4. This framework is not a standard NMF. It is important to provide more information in the Methods section to explain how different template matrices are learned with the same weight matrix. Specifically, the process of "unraveling the 3D input matrices into 2D vectors suitable for NMF" needs a detailed explanation to enhance reproducibility and understanding.

5. From a computational perspective, the novelty of the proposed framework appears limited.

Reviewer #3: In the article "ChromaFactor: deconvolution of single-molecule chromatin organization with non-negative matrix factorization," Laura M. Gunsalus, Michael J. Keiser, and Katherine S. Pollard consider the topic of deconvolution of single-molecule chromatin organization with non-negative matrix factorization.

The authors address the problem that analysis of single-molecule data is hampered by extreme yet inherent heterogeneity, making it challenging to determine the contributions of individual chromatin fibers to bulk trends. They mention that several computational methods have been developed in response to emerging single-cell imaging and high-throughput sequencing techniques to measure chromatin conformation. They claim that these works did not yet connect the behavior of individual cells to populations of similar conformations that are transcriptionally on or off.

The authors provide a computational approach based on non-negative matrix factorization that deconvolves single-molecule chromatin organization datasets into components and identifies subpopulations that correlate with cellular phenotypes.

It is not clear whether ChromoFactor enhances the large body of existing literature on single-cell chromatin architecture. A more thorough summary of the literature needs to be provided and compared and contrasted with ChromoFactor.

It isn't easy to assess the significance of the method in the context of 3D genome biology research since only one biological example involving a single locus was provided and the utility of the reported result is not clear.

**Have the authors made all data and (if applicable) computational code underlying the findings in their manuscript fully available?**

Reviewer #1: Yes

Reviewer #2: Yes

Reviewer #3: Yes

PLOS authors have the option to publish the peer review history of their article (what does this mean? ). If published, this will include your full peer review and any attached files.

**Do you want your identity to be public for this peer review?** For information about this choice, including consent withdrawal, please see our Privacy Policy .

Reviewer #1: No

Reviewer #2: No

Reviewer #3: No
---

## [Decision Letter · Decision Letter 1]

2 Feb 2025

Dear Dr. Pollard,

We are pleased to inform you that your manuscript 'ChromaFactor: deconvolution of single-molecule chromatin organization with non-negative matrix factorization' has been provisionally accepted for publication in PLOS Computational Biology.

Best regards,

Jie Liu

Academic Editor

PLOS Computational Biology

Jian Ma

Section Editor

PLOS Computational Biology

Reviewer's Responses to Questions

**Comments to the Authors:**

Reviewer #1: I'd like to thank the authors for their thorough revisions and for addressing my concerns. Their responses and changes have greatly improved the manuscript.

Reviewer #2: The authors have addressed the comments in the revised version of the manuscript. Therefore, I have no further comments.

Reviewer #3: The authors present a computational method that uses non-negative matrix factorization to analyze single-molecule chromatin organization datasets, breaking them down into components and identifying subpopulations that correlate with cellular phenotypes.

The authors have revised the manuscript to clarify how ChromoFactor enhances the large body of existing literature on single-cell chromatin architecture. They have also provided an improved literature summary and compared previous work to ChromoFactor.

The authors have added more than one biological example, making it easier to assess the method's significance in the context of 3D genome biology research. The utility of the reported results is clearer.

**Have the authors made all data and (if applicable) computational code underlying the findings in their manuscript fully available?**

Reviewer #1: Yes

Reviewer #2: Yes

Reviewer #3: Yes

PLOS authors have the option to publish the peer review history of their article (what does this mean? ). If published, this will include your full peer review and any attached files.

**Do you want your identity to be public for this peer review?** For information about this choice, including consent withdrawal, please see our Privacy Policy .

Reviewer #1: No

Reviewer #2: No

Reviewer #3: No

---

## [Editor Report · Acceptance letter]

PCOMPBIOL-D-24-00851R1

ChromaFactor: deconvolution of single-molecule chromatin organization with non-negative matrix factorization

Dear Dr Pollard,

I am pleased to inform you that your manuscript has been formally accepted for publication in PLOS Computational Biology. Your manuscript is now with our production department and you will be notified of the publication date in due course.

With kind regards,

Anita Estes
